# $E^2$: Easy Regional Contrastive Learning of Expressive Fashion Representations

**Daiqing Qi**
University of Virginia
Charlottesville, VA 22904
daiqing.qi@virginia.edu

**Handong Zhao**
Adobe Research
San Jose, CA 95110
hazhao@adobe.com

**Sheng Li**
University of Virginia
Charlottesville, VA 22904
shengli@virginia.edu

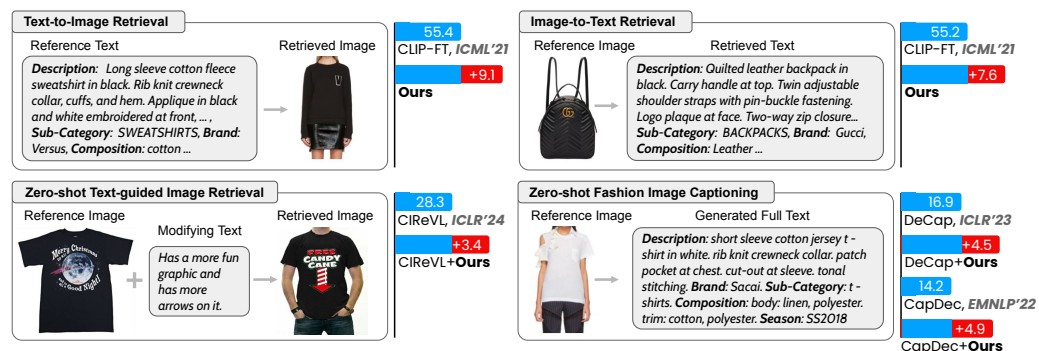

Figure 1: Model performance on downstream tasks. The first row shows the direct application of our model in cross-modal retrieval. The second row shows that, when replacing the CLIP encoders with ours in existing SOTA models in *zero-shot* text-guided image retrieval and fashion image captioning, their results can be further improved notably. (We show R@1 results for cross-modal retrieval, average R@10 for text-guided image retrieval, and B@4 for image captioning. )

## Abstract

When learning vision-language models (VLM) for the fashion domain, most existing works design new architectures from vanilla BERT with additional objectives, or perform dense multi-task learning with fashion-specific tasks. Though progress has been made, their architecture or objectives are often intricate and the extendibility is limited. By contrast, with simple architecture (comprising only two unimodal encoders) and just the contrastive objective, popular pre-trained VL models (e.g., CLIP) achieve superior performance in general domains, which are further easily extended to downstream tasks. However, inheriting such benefits of CLIP in the fashion domain is non-trivial in the presence of the notable domain gap. Empirically, we find that directly finetuning on fashion data leads CLIP to frequently ignore minor yet important details such as logos and composition, which are critical in fashion tasks such as retrieval and captioning. In this work, to maintain CLIP's simple architecture and objective while explicitly attending to fashion details, we propose $E^2$: **E**asy Regional Contrastive Learning of **E**xpressive Fashion Representations. $E^2$ introduces only a few selection tokens and fusion blocks (just 1.9% additional parameters in total) with only contrastive losses. Despite lightweight, **in our primary focus, cross-modal retrieval**, $E^2$ notably outperforms existing fashion VLMs with various fashion-specific objectives. Moreover, thanks to CLIP's widespread use in downstream tasks in general domains (e.g., zero-shot composed image retrieval and image captioning), our model can easily extend these models from general domain to the fashion domain with notable improvement (Fig. 1). To conduct a comprehensive evaluation, we further collect data from Amazon Reviews to build a new dataset (Fig. 4) for cross-modal retrieval in the fashion domain.

38th Conference on Neural Information Processing Systems (NeurIPS 2024).

*General Domain*                *Fashion Domain*

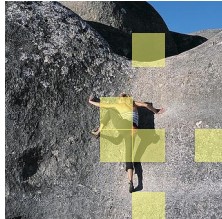 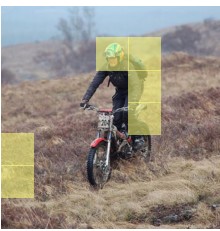 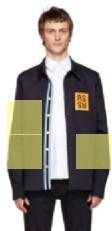 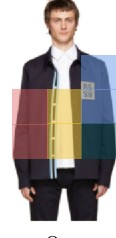

**Description**: Long sleeve denim shirt in dark navy. Spread collar. Button closure at front featuring rib knit trim striped in blue, white, and black. Patch pocket and leather logo patch in orange at chest. Single-button barrel cuffs. Silver-tone hardware. Tonal stitching.

**Category**: SHIRTS.
**Brand**: Raf Simons.
**Season**: FW2016 (Fall/Winter).
**Composition**: 96% cotton, 4% polyurethane.

A girl climbs a huge rock.    A man riding a bike on a hill.    CLIP-FT    Ours

Figure 2: Illustration of domain gap and attention results of CLIP and $E^2$. General domain data often consist of a short caption which describes *a few* objects in an image, while fashion data come with description and meta information (tag entities) of a *single* product. **CLIP**: Image Tokens with maximum attention values (in each attention head) with the global token are marked yellow. $E^2$: Selected image tokens by selection tokens during the second stage are colored. Blue: Brand. Orange: Season, Red: Sub-category. Green: Composition.

# 1 Introduction

There has been a long research line for vision-language learning in general domain [15, 34, 19, 23, 26, 40, 52, 53]. Recently, cross-modal retrieval in fashion domain is receiving increasing attention [10, 27, 12, 56, 13, 4, 38, 22, 14, 12]. Most existing fashion vision-language models (VLM) [12, 38, 14, 10, 12, 56] design new vision-language models based on vanilla BERT [6].

Existing studies commonly train BERT-based models with Masked Language Modeling (MLM), (Image-Text Matching) ITM, Image-Text Contrastive Learning (ITC), or Masked Image Modeling (MIM) to their specific architectures. Various techniques tailored to fashion domain have been proposed, including learning extra fashion- specific tasks, new attention mechanism [38], or additional modules for fashion feature learning [14]. FashionSAP [14] and FAME-ViL [13] perform a fashion-specific multi-task learning with various fashion tasks in addition to cross-modal retrieval, such as category recognition. However, the learning of existing BERT-based or multi-task fashion VLMs is often intricate with their complex architecture or additional fashion-specific objectives. By contrast, with simple architecture that comprises only two unimodal transformer encoders, and a single contrastive learning objective, contrastive language-image pre-training models such as CLIP [34] exhibit outstanding performance from cross-modal retrieval to a wide range of downstream tasks [24, 42, 20, 33]. Motivated by the simplicity and effectiveness of CLIP, we look forward to learning a model that inherits such benefits of CLIP in the fashion domain, which could be simple, lightweight while highly effective. While directly finetuning CLIP (FashionCLIP [4]) is an intuitive solution, it is deficient in presence of the notable domain shift [55]. Consequently, Ma et al., [27] uses CLIP as backbone and improves it by using additional text encoders to mitigate the word ambiguity in fashion language. However, they ignore the uniqueness of visual learning in fashion domain.

Different from data in general domain, product images and descriptions in fashion domain are unique in several aspects. As illustrated in Fig. 2, in general domain, an image contains only a few distinctive objects, and text descriptions are more concise and general. However, in fashion domain, a product image usually includes only one foreground object but with rich details. Besides, the fashion text often provides a group of metadata (tag entities) [16, 36], such as Composition, Brand, Description, Sub-category, etc. When directly finetuning CLIP on fashion data, we find it tends to give more attention to regions that are closer to a global view, e.g., dark regions in dark clothing (Fig. 2 and 14). Consequently, it misses details associated with tag entities such as composition and logo, which represent more of a local view and are critical for distinguishing visually similar items.

To quantify the capability of visual representation in recognition of product details, we design an exploratory entity classification task based on *linear probing*: fitting a linear classifier on image embeddings from different models respectively, with tag entities used as labels. Results in Fig. 3 show that, the classifier which learns from fine-tuned CLIP (CLIP-FT) embeddings are less effective, **indicating that CLIP-FT embeddings contain less entity-related information, i.e., CLIP-FT is ineffective in extracting entity-related information from fashion images.** Details are available in Appendix E, where we provide a more comprehensive analysis.

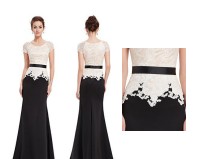

**①** *AmazonFashion (Ours)*

**Description**: Ever-Pretty Womens Elegant Floor Length Mother Of The Groom Dress 14 US Black and White

**Brand**: Ever-Pretty

**Category**: Clothing

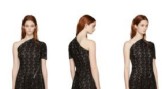

**②** *FashionGen*

**Description**: Short-sleeve lace dress in black. Single-shoulder. Zipper accent at front in gold-tone. Asymmetric seam at waist. Silk lining in beige. Tonal stitching. **Category**: LONG DRESSES **Brand**: Stella McCartney **Season**: FW2016 **Composition**: Body: 70% cotton, 30% nylon. Lining: 100% silk.

*Dataset Statistics*

| Dataset | ① (Ours) | ② |
|---|---|---|
| **Image-Text Paris** | 1.3M | 29.6K |
| **Brand Count** | 38211 | 570 |
| **Product Count** | 544713 | 67666 |
| **Avg. Desc. Length** | 11.43 | 28.92 |

Figure 4: Comparison of FashionGen [36] and AmazonFashion (Ours), which is notably different in size, diversity, language style and image scope (close-up shots of products are included). We compare it with more datasets [7, 51], and show more details (e.g., clustering [52]) in Appendix D.

Then it is natural to ask: is it possible to have a CLIP image encoder that learns richer visual representations towards tag entities without eroding its learnt knowledge?

Towards this end, we propose our model: $E^2$: **E**asy Regional Contrastive Learning of **E**xpressive Fashion Representations. Without modifying the Vision Transformer (ViT) [9] structure of the image encoder, $E^2$ learns richer representations with the

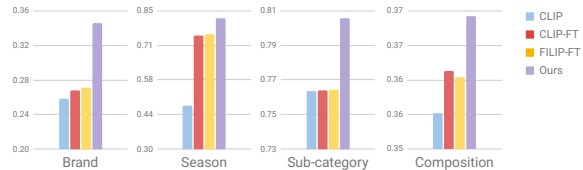

Figure 3: Linear Probing Results on FashionGen. The more informative embeddings are, the higher accuracy a classifier obtains.

guidance of region contrastive loss. Meanwhile, it explicitly pays more attention to details in images by selecting most relevant image patch tokens that contain rich detailed information and fusing them with selection tokens during the forward propagation in ViT. Our contributions are follows:

**(1)** We first reveal that the image encoder of pre-trained CLIP is biased towards visually dominant regions of a product and consequently ignores small but critical details *when directly fine-tuned on fashion domain*. Furthermore, we present the first simple yet effective solution to this problem by allowing the model to learn more fine-grained visual representations towards tag entities. **(2)** Motivated by the observation in (1), we propose a concise regional contrastive learning framework comprising only two unimodal encoders and just contrastive learning objectives, while it learns richer visual representations. **(3)** We conduct extensive experiments with existing benchmark datasets, including our new benchmark dataset (Fig. 4) for fashion *cross-modal retrieval*, which differs from the popular benchmark FashionGen [36] in notably larger size, a wider variety of brands and products, more concise and general descriptions, and more diverse image scopes, making it more challenging and more practical. **(4)** Inheriting the benefits of the widely used CLIP, our model can easily extend models from general domain to the fashion domain with notable improvement (Fig 1) in downstream tasks (e.g., zero-shot composed image retrieval and image captioning).

## 2    Related Work

**Text and Image Matching.** Text and image matching aims to semantically align the text and image. Various BERT-based vision-language models (VLM) [40, 26, 41, 25, 14, 12, 10] are proposed towards this end. Different from previous studies, CLIP [34] introduces a large-scale vision language pre-training framework, which learns from 400 million image-text pairs with contrastive learning. Despite of the simplicity of its structure, CLIP is surprisingly effective in visual-language learning and various downstream tasks [24, 42, 20].

**Fashion Cross-modal Retrieval.** Different from general domain, the fashion data involves large amount of domain-specific information with richer details than data from general domain, such as the brand, material, texture, composition, various of minute design differences, etc. Besides, the fashion text is composed of description and a group of tag entities (meta information), which describes the meta information of products, such as its sub-category, season, brand, to name a few. Wang et al. [45] enhances the task with knowledge graph [17, 37, 31]. A line of research pre-trains BERT-based models [10, 56, 12, 14, 12] that are tailored for fashion data. Fame-ViL [13] and FashionSAP [14] further formulate a multi-task learning framework with extra fashion-specific objectives. Different from existing BERT-based models, Chia et al. [4] and Ma et al. [27] are built upon powerful CLIP. While Chia et al. [4] directly finetune CLIP for continual learning [32] with fashion data, Ma et al. [27]

improve its language learning with additional text encoders to diminish word ambiguity. However, they do not consider the visual discrepancy. To better learn from fashion images, which contain richer detailed information, we improve CLIP from the perspective of visual learning, enabling its image encoder to learn more fine-grained representations for better image-text alignment.

**Zero-shot Text-guided Image Retrieval.** In Text-guided Image Retrieval (TGIR), also known as compositional image retrieval (CIR), users perform interactive dialogue to refine a given query image toward retrieving specific items. Classic models often employ custom models that project text-image pairs into a common embedding space. With the advance of VL foundation models (e.g., CLIP), interest in CIR has surged, especially in zero-shot settings without task-specific models [20]. We show that when combined with $E^2$, which learns more fine-grained representations towards fashion product details, their performances can be further improved.

**Zero-shot Image Captioning.** Zero-shot captioning [3, 46, 1, 42, 50, 24, 30] aims to generate image/video captions without human-annotated data. Different from above works, built upon CLIP, DeCap [24] and CapDec [30] use text-only data to train a decoder from scratch. We show that when combined with $E^2$, which learns more fine-grained representations towards fashion product details, their performances can be further improved.

We provide a detailed discussion on our innovation compared with existing works in Appendix G.

## 3 Methodology

### 3.1 Contrastive Language-Image Pre-training

Instead of learning from predicting a fixed set of predetermined object categories, i.e., the classification task, CLIP (Contrastive Language-Image Pre-training) [34] directly learns visual representations from raw text, and it is trained on 400 million image-text pairs with contrastive learning. Specifically, given a batch of $N$ image-text pairs $\{(\boldsymbol{I}_i, \boldsymbol{T}_i)\}_{i=1}^N$, images and texts are encoded as $d$-dimensional embeddings by the image encoder $h^I(\cdot)$ and the text encoder $h^T(\cdot)$. Denote the image embedding and text embedding as $\boldsymbol{z}_i^I$ and $\boldsymbol{z}_i^T$, respectively. During training, CLIP learns image-text matching from of $N \times N$ possible combinations by maximizing the similarity scores of $N$ matched pairs while minimizing the scores of the rest $N^2 - N$ mismatched pairs. The cosine similarity score of an image-text pair is calculated as $\boldsymbol{z}_i^I \odot \boldsymbol{z}_i^T$, $(i, j \in \{1, 2, ..., N\})$. In practice, CLIP optimizes a cross-entropy loss over the $N \times N$ similarity scores matrix, namely contrastive loss, denoting as:

$$\mathcal{L}_{contra}(\{(\boldsymbol{z}_i^I, \boldsymbol{z}_i^T)\}_{i=1}^N) = \mathcal{L}_{I2T} + \mathcal{L}_{T2I}, \tag{1}$$

where $\mathcal{L}_{I2T} = -\frac{1}{N} \sum_{i=1}^N \log \frac{\exp(\boldsymbol{z}_i^I \cdot \boldsymbol{z}_i^T / \tau)}{\sum_{j=1}^N \exp(\boldsymbol{z}_i^I \cdot \boldsymbol{z}_j^T / \tau)}$, and $\mathcal{L}_{T2I} = -\frac{1}{N} \sum_{i=1}^N \log \frac{\exp(\boldsymbol{z}_i^T \cdot \boldsymbol{z}_i^I / \tau)}{\sum_{j=1}^N \exp(\boldsymbol{z}_i^T \cdot \boldsymbol{z}_j^I / \tau)}$,

$\tau$ is the temperature scalar. CLIP proves its effectiveness on various tasks in general domain. However, when adapting it to fashion domain, it is particularly difficult for CLIP to effectively learn from the fashion data due to its uniqueness in richer details and more compact layout. To overcome the challenges, we make full use of tag entities and learn more fine-grained representations with explicit (1) token fusion and selection, and (2) region contrastive learning.

### 3.2 Regional Contrastive Learning of Fashion Representations

We present details of our framework, **E**asy Regional Contrastive Learning of **E**xpressive Fashion Representations ($E^2$) in Fig. 5. The core idea of $E^2$ is "Easy" and "Expressive": it inherits the (1) simple design of CLIP with only a few inserted fusion blocks and selection tokens in the vision encoder, and (2) the simple learning objective: only contrastive learning objectives are used. Yet $E^2$ is more effective than existing large fashion VLMs with various objectives.

In the following, we first describe the overall framework of $E^2$, and then introduce the selection tokens, which are key elements for our (1) token fusion and selection, and (2) region contrastive learning. After that, we explain each component and finally summarize the whole training process.

**Framework.** We build $E^2$ upon CLIP [34], where we keep its text encoder unchanged and facilitate its image encoder with proposed selection tokens, fusion blocks and region contrastive loss. Similar to CLIP, given an image-text pair, $E^2$ learns one global embedding for each for the calculation of similarities for contrastive learning.

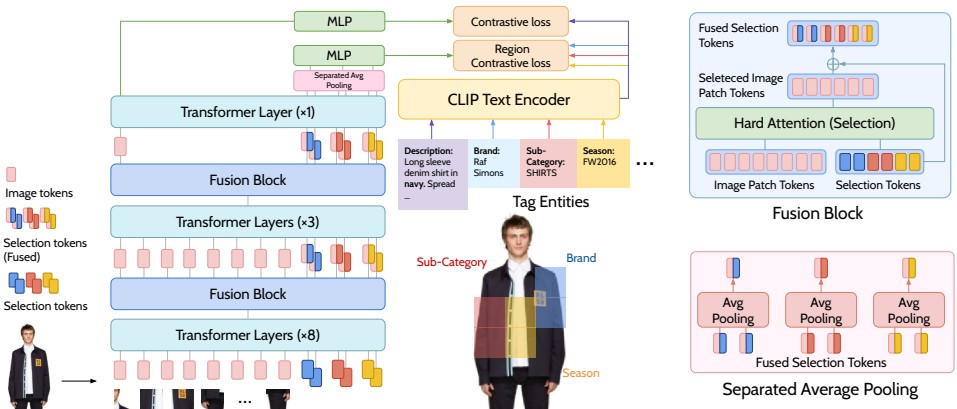

Figure 5: (a) **Framework.** $E^2$ contains an image encoder and a text encoder. The image encoder consists of CLIP transformer layers with inserted fusion blocks, where selection tokens update themselves with most relevant image tokens. After the second stage of token fusion, only the global image token and selection tokens are kept as input for the last transformer layer. Selection tokens further learn entity-specific knowledge with region contrastive loss. (b) **The Architecture of Fusion Block.** In fusion block, each selection token selects one most relevant image patch tokens and update itself with the averaged embedding of itself and the selected token. (c) **Visualization**. Logo is covered by blue masks. Front zipper area indicates its season (Fall/Winter 2016) and left front area with sleeves suggest the sub-category (shirts). *Note that the figure is only for illustration, in experiments, we consider $C = 4$ tag entities and assign $S = 2$ selections for each.*

**Selection Tokens.** Given an image, it is first split into $P$ non-overlapping patches, which are linearly projected into a latent space, denoted as $\{\boldsymbol{p}_i\}_{i=1}^P$. In addition to $P$ image tokens, we further propose a set of *selection tokens* categorized into $C$ categories, with $S$ tokens for each sub-category, denoted as $\{\boldsymbol{s}_i\}_{i=1}^{C \times S}$. For instance, for the image encoder in Fig. 5, $C = 3$ and $S = 2$, The two blue, red and yellow selection tokens are associated with the tag entity *Brand*, *Sub-category* and *Season*, respectively. To capture detailed information in image patches, each selection token updates itself with one most relevant image token in fusion block. Note that selection tokens are supervised by the region contrastive loss. We will discuss it later in this section.

**Multi-Stage Image Token Fusion and Selection.** To learn more fine-grained visual representations with the help of tag entities in fashion language, we use a group of selection tokens to iteratively select most informative images tokens and merge themselves with the selected ones. In this way, selection tokens contain rich information about the details of their associated tag entities. For more fine-grained interactions, we drop less informative image tokens and only keep the global embedding and selection tokens as the input of the *last* transformer layer. As shown in Fig. 5, we perform a multi-stage token fusion and selection to obtain the final global embedding. Given an image, we first obtain $P$ image tokens $\{\boldsymbol{p}_i\}_{i=1}^P$. Then they are concatenated with a set of selection tokens $\{\boldsymbol{s}_i\}_{i=1}^{C \times S}$ and input to the image encoder, where we perform a multi-stage token fusion and selection. In each stage, input tokens sequentially go through a few transformer layers and a fusion block. Formally, suppose there are $L$ stages, and during the $l$-th stage, we denote the input tokens as $\{\boldsymbol{p}_i^l\}_{i=1}^P$ and $\{\boldsymbol{s}_i^l\}_{i=1}^{C \times S}$. The information propagation with each group of transformer layers is performed:

$$\{\hat{\boldsymbol{p}}_i^l\}_{i=1}^P, \{\hat{\boldsymbol{s}}_i^l\}_{i=1}^{C \times S} = \text{Transformer}([\{\boldsymbol{p}_i^l\}_{i=1}^P; \{\boldsymbol{s}_i^l\}_{i=1}^{C \times S}]), \tag{2}$$

where $[;]$ means concatenation. Then the obtained image and selection tokens are fed to the fusion block, where each selection token selects one most related image token and updates the embedding of itself with the selected image token:

$$\{\boldsymbol{p}_i^{l+1}\}_{i=1}^P, \{\boldsymbol{s}_i^{l+1}\}_{i=1}^{C \times S} = \text{FusionBlock}([\{\hat{\boldsymbol{p}}_i^l\}_{i=1}^P; \{\hat{\boldsymbol{s}}_i^l\}_{i=1}^{C \times S}]), \tag{3}$$

After that, the output tokens serve as input tokens for stage $l+1$ if it is not the last stage. After obtaining the output tokens from the last stage $L$ (via Eq. 3 with $l=L$), we only keep the global embedding $\boldsymbol{p}_0^{L+1}$ and selection tokens $\{\boldsymbol{s}_i^{L+1}\}_{i=1}^{C \times S}$ as the input to the last transformer layer. To enrich the global embedding with fine-grained features related to given tag entities, we enforce it to

focus on interacting with informative selection tokens by dropping less relevant image tokens to avoid their distractions. This step is critical for the image encoder to effectively learn more fine-grained visual representations with tag entities. As shown in Fig. 5(a), the last transformer layer is applied on the kept tokens to obtain $\hat{p}_0^{L+1}$. Finally, the global image embedding $z^I$ is obtained by applying a MLP to it:

$$\hat{p}_0^{L+1} = \text{Transformer}([p_0^{L+1}; \{s_i^{L+1}\}_{i=1}^{C \times S}]), \text{ and } z^I = \text{MLP}(\hat{p}_0^{L+1}). \tag{4}$$

**Fusion Block.** In fusion blocks, selection tokens are fused with their most relevant image patch tokens to enrich themselves with entity-specific (e.g., brand, composition, etc.) visual information. Given a group of image patch tokens $\{p_i^l\}_{i=1}^P$, denoting as a matrix $P^l$, and a group of selection tokens $\{s_i^l\}_{i=1}^{C \times S}$, denoting as a matrix $S^l$, we use selection tokens as queries to select the most relevant image tokens, which we call *hard attention*. For each selection token, it is updated by averaging itself with the selected image patch token. Specifically, denoting $Q^l$, $K^l$ and $V^l$ are linear projections of $S^l$ and $P^l$ respectively: $Q^l = W_q S^l$, $K^l = W_k P^l$ and $V^l = W_v P^l$. Attention weight matrix $A^l$ is calculated by:

$$A_{i,j}^l = \frac{\exp\left(Q_{i,:}^l K_{:,j}^l + \gamma_i\right)}{\sum_{k=1}^P \exp\left(Q_{i,:}^l \cdot K_{:,k}^l + \gamma_k\right)} \tag{5}$$

where $\gamma_i$ is the i.i.d random sample drawn from the $Gumbel(0,1)$ distribution. To explicitly select one most similar image patch token for each selection token, we reformulate the attention weight matrix by turning each row $A_{i,:}$ into a one-hot representation with assigning one to the term with highest similarity score and zero to the rest. As the $argmax$ operation is not differentiable, we use the gumbel-softmax and straight-through trick [8, 44, 48]:

$$\hat{A}_{i,:}^l = \text{one-hot}(\underset{j}{\arg\max} A_{i,j}^l) + A_{i,:}^l - \text{stop}(A_{i,:}^l), \tag{6}$$

where the operator $\text{stop}$ stops the gradients propagation. With one-hot vectors in the attention weight matrix $A^l$, each selection token can pick up one corresponding image patch token and update itself with the selected one via:

$$S_{i,:}^{l+1} = S_{i,:}^l + V_{i,:}^l A^l. \tag{7}$$

In fusion blocks of different stages, selection tokens constantly select most relevant image tokens explicitly and enrich the representations of themselves with selected token embeddings for better selection and fusion in the next stage.

At the last stage, the output of the fusion block only contains selection tokens, which already contain rich fine-grained information about associated tag entities (i.e., *brand*, *composition*, *season* and *categories*, etc.). They are input to the last transformation layer with the global image token, so that the global token can effectively interact with them without distractions from irreverent patch tokens. In this way, the global token better captures details of fashion images.

**Region Contrastive Learning with Selection Tokens.** As each selection token is associated with a tag entity and it aims to select most relevant image tokens in fusion blocks, selection tokens are further supervised with the region contrastive loss. Specifically, assume we have a group of selection tokens $\{s_i\}_{i=1}^{C \times S}$, which are categorized into C categories, with $S$ tokens for each sub-category. For better illustration in this subsection, we reformulate them as $\{s_s^c\}$ where $c \in \{$'brand', 'composition', ...$\}$ (C categories in total) and $s \in \{1, 2, ..., S\}$. After going through the last transformer layer, each set of selection tokens $\{s_s^c\}_{s=1}^S$ with the same associated sub-category $c$ is fed to its corresponding average pooling layer (Fig. 5 right), so that selection tokens from the same sub-category are formed to a more comprehensive and informative single pooled embedding $s_{final}^c$ as:

$$s_{final}^c = \text{AvgPool}(\{s_s^c\}_{s=1}^S), \ c \in \{\text{'brand'}, ...\}. \tag{8}$$

Similar to the contrastive learning process with global image embeddings and global text embeddings introduced in section 3.1, each $s_{final}^c$ learns to match the corresponding ground truth tag entity $z^{T(c)}$ via the region contrastive loss by replacing the global image embedding with the corresponding pooled selection token in Eq. 1:

$$\mathcal{L}_{region}(\{(z_i^{T(c)}, s_{final_i}^c)\}_{i=1}^N) = \mathcal{L}_{I2T} + \mathcal{L}_{T2I}. \tag{9}$$

The region contrastive loss explicitly aligns each $s_{final}^c$ with the corresponding ground truth tag entity $z^{T(c)}$ for sub-category $c$. With region contrastive loss, pooled selection tokens are distinguishable

| Model | Image to Text | | | Text to Image | | | SumR | MeanR@1 |
|---|---|---|---|---|---|---|---|---|
| | R@1 | R@5 | R@10 | R@1 | R@5 | R@10 | | |
| ALBEF [22] | 41.7 | - | - | 51.0 | - | - | - | 46.2 |
| SyncMask [38] | 55.4 | - | - | 64.0 | - | - | - | 59.7 |
| FashionSAP [14] | 54.3 | 77.3 | 83.2 | 62.8 | 83.9 | 90.6 | 451.8 | 58.5 |
| FashionViL [12] | 42.8 | 71.5 | 80.6 | 51.3 | 75.4 | 84.7 | 406.5 | 47.1 |
| FashionCLIP [4] | 54.3 | 85.5 | 92.8 | 54.4 | 85.9 | 92.3 | 465.2 | 54.4 |
| FILIP-FT* [49] | 55.5 | 86.1 | 93.2 | 55.8 | 86.5 | 93.1 | 470.2 | 55.7 |
| CLIP-FT [34] | 55.2 | 85.7 | 92.9 | 55.4 | 86.2 | 92.9 | 468.2 | 55.3 |
| Ours | **62.8** | **89.7** | **95.3** | **64.5** | **90.1** | **95.5** | **497.9** | **63.7** |

| Model | Image to Text | | | Text to Image | | | SumR |
|---|---|---|---|---|---|---|---|
| | R@1 | R@5 | R@10 | R@1 | R@5 | R@10 | |
| CLIP-VPT [18] | 15.7 | 43.1 | 60.0 | 16.6 | 43.3 | 59.4 | 238.1 |
| FashionBert [10] | 23.9 | 46.3 | 52.1 | 26.7 | 46.4 | 55.7 | 251.1 |
| KaleidoBert [56] | 27.9 | 60.0 | 68.3 | 33.8 | 60.6 | 68.5 | 319.1 |
| FaD-VLP [28] | 64.3 | 86.8 | 93.5 | 58.7 | 84.5 | 91.6 | 479.7 |
| FAME-ViL [13] | 65.9 | 91.9 | 97.2 | 62.9 | 87.4 | 93.5 | 498.8 |
| FILIP-FT* [34] | 68.1 | 95.0 | 98.5 | 65.3 | 93.6 | 98.2 | 518.7 |
| CLIP-FT [34] | 67.4 | 94.1 | 98.2 | 65.0 | 93.2 | 98.0 | 515.9 |
| Ours | **76.2** | **96.8** | **99.1** | **73.1** | **95.3** | **99.1** | **538.6** |

Table 1: Retrieval performances (full evaluation) on Fashion-Gen. (*) denotes results from our implementation. As the code and pre-trained model of FILIP are not released yet, we implemented the model and initialize it with pre-trained CLIP weights.

Table 2: Retrieval performances (sample-100 evaluation) on Fashion-Gen. (*) denotes results from our implementation.

towards tag entities, which means learnable selection tokens can effectively select and fuse themselves with tag entity-specific image patch tokens given fashion images. Finally, the global image embedding learns more fine-grained representation by: (1) interactions with these selection tokens in transformer layers, and (2) explicit token selection process before the last transformer layer, which filters less relevant patch tokens and helps the global embedding to concentrate more on certain regions.

**Remark.** Some semantic segmentation models, e.g., Seg [39] and GroupViT [48] also involve additional tokens. But the usage is notably different. As their motivation is to group image tokens to larger objects, each image token is assigned to an additional token, where softmax is applied to ***additional tokens***. Because our motivation is to pay more attention to details of an object, each additional (selection) token selects a most relevant image token, where softmax is applied to ***image tokens***. We carefully discuss the differences in detail in Appendix H.

# 4 Experiments

**Datasets.** For a fair comparison, we first evaluate our model on the benchmark dataset Fashion-Gen [36], following existing works [10, 56, 12, 27]. Besides, we also collect text descriptions and product images in fashion domain from Amazon Reviews [29] and build a large-scale fashion dataset which contains 1.3M image-text pairs, where we use 910K and 390K pairs for training and test, respectively. It is a more challenging dataset as the text descriptions are briefer and more general than FashionGen. We refer this dataset as AmazonFashion. More details are available in Appendix D.

**Settings.** $E^2$ is initialized from the pre-trained CLIP [34] (ViT-B-32 [9]) We use two selection tokens for each tag entity ($S = 2$), as Fig. 5 shows. We present the validation of this choice in later experiments. More detailed configuration and hyper-parameter setting are available in Appendix A.

**Evaluation.** Our model is evaluated in various downstream tasks. For retrieval tasks, we perform two kinds of evaluations. The positive candidate is the ground truth item, while the negative candidates are either 100 randomly sampled unmatched items (referred as sample-100 evaluation) or all unmatched items in the dataset (full evaluation). Following [10, 27], the evaluation metrics are Rank@1, Rank@5, Rank@10 and $SumR$=100(Rank@1 + Rank@5 + Rank@10).

**Note** that full evaluation, which is suggested by latest works [27, 12], is more challenging and practical and consistent with practical retrieval tasks. We compare with baselines with full evaluation unless they are not compatible with full evaluation, where we run Sample-100 evaluation instead.

## 4.1 Comparison with State-of-the-Art

**Cross Model Retrieval.** For full evaluation on FashionGen, we compare our model with existing fashion VLMs and finetuned CLIP-based models. To compare with more baselines which are not compatible with full evaluation, following [10, 56], we also do a Sample-100 evaluation. Because CLIP-FT and FILIP-FT outperform other baselines by a large margin, we compare $E^2$ with the two closest baselines on AmazonFashion with full evaluation. Table 1 shows $E^2$ notably outperforms other baselines in full evaluation, manifesting the effectiveness of paying more attention to certain regions

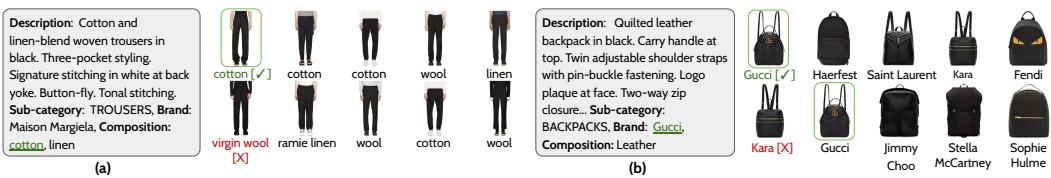

Figure 6: **T2I retrieval examples.** For each example, the query text is displayed on the left. The first row is the top-5 retrieval result by $E^2$ and the second row is the result by CLIP-FT. Ground truth is boxed in green. For each product image, its Composition (a) or Brand (b) is marked below. For top-1 retrieved item, we mark its tag entity red if it is not consistent with the query text.

that contain rich information about tag entities. Table 2 shows our model also outperforms baselines in sample-100 evaluation by a large margin. On the more challenging large-scale AmazonFashion dataset, $E^2$ consistently achieves better performance than competing baselines (Table 3).

In Fig. 6, we present examples of retrieval results by $E^2$ and CLIP to illustrate in which cases CLIP fails to differentiate minor differences towards visually similar products. In Fig. 6(a), both CLIP and $E^2$ retrieved visually similar black trousers. While $E^2$ ranks the ground truth sample first, CLIP ranks visually similar product higher without considering its composition. . Fig. 6(b) shows $E^2$ ranks the *Gucci* backpack first by recognizing its logo at face while CLIP ranks the *Kara* backpack first incorrectly. More visualizations are available in Appendix B.

**Zero-shot Image Captioning (ZS-IC).** Built upon CLIP, $E^2$ can be easily extended to popular SOTA CLIP-based zero-shot image captioning models [24, 30]. DeCap [24] and CapDec [30] share similar high-level ideas: during training, a language decoder is trained to reconstruct the text input, where the CLIP text encoder serves as the encoder. During inference, with the CLIP image encoder, a given image is first encoded to a CLIP feature, which is later fed to the pre-trained decoder to generate captions. We train the decoder on text-only data from FashionGen but with our $E^2$ as the backbone encoder instead of CLIP. Results are shown in Tab. 4. As learnt features by $E^2$ contain more fine-grained information towards product specifications, generated captions with $E^2$ better matches ground truth captions, leading to high scores. We show the example in Fig 7. Vanilla DeCap tends to make more mistakes towards details, such as brand, composition. While our improved DeCap with $E^2$ constantly yields better results.

**Zero-shot Text-guided Image Retrieval (ZS-TGIR).** CIReVL [20] exploits pre-trained vision-language models (CLIP) alone with an LLM for ZS-TGIR without training. Similar to the case in ZS-IC, when replacing the CLIP encoders with $E^2$, CIReVL is easily extended to fashion domain, consequently the model performance notably improves in Tab. 8.

**Remark.** EI-CLIP [27] and fine-grained CLIPs [49] are not necessarily our baselines, as our focuses are different and orthogonal to each other. Still we have a comparison to demonstrate the uniqueness and advantages of our model. Detailed results and discussions are presented in Appendix H.

### 4.2 Ablation and Further Analysis

To study the effectiveness of each module and how much each group of selection tokens contribute to the our model, we design ablation studies from the two perspective: architecture and selection tokens. **Architecture.** We first ablate fusion blocks, where the input and output of each transformer layer are

| Model | Image to Text | | | Text to Image | | | *SumR* |
|---|---|---|---|---|---|---|---|
| | R@1 | R@5 | R@10 | R@1 | R@5 | R@10 | |
| FILIP-FT* [49] | 6.2 | 17.8 | 25.8 | 6.2 | 18.1 | 25.9 | 100.0 |
| CLIP-FT [34] | 6.1 | 17.7 | 25.8 | 6.2 | 17.9 | 25.8 | 99.5 |
| $E^2$ (Ours) | **7.5** | **20.3** | **28.4** | **7.4** | **20.2** | **28.4** | **112.2** |

Table 3: Retrieval performances (full evaluation) on AmazonFashion. (*) denotes results from our implementation.

| Model | B@4 | C | M | R | S |
|---|---|---|---|---|---|
| CapDec [30], EMNLP'22 | 14.23 | 5.14 | 14.98 | 15.02 | 17.87 |
| - CapDec w/ $E^2$ | 19.13 | 10.13 | 18.91 | 18.77 | 23.82 |
| DeCap [24], ICLR'23 | 16.88 | 6.61 | 16.21 | 16.41 | 19.42 |
| - DeCap w/ $E^2$ | **21.35** | **12.18** | **20.93** | **20.60** | **25.03** |

Table 4: **Zero-shot** image captioning results on FashionGen with BLEU@4 (B), CIDEr (C), METEOR (M), ROUGE (R).

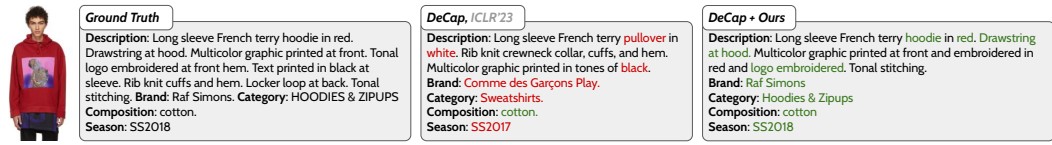

Figure 7: **Zero-shot Image Captioning examples.** We mark the wrong captions red, and mark a caption green if it *exactly* matches the ground truth.

| Method | Shirt | | Dress | | Toptee | | Average | |
|---|---|---|---|---|---|---|---|---|
| | R@10 | R@50 | R@10 | R@50 | R@10 | R@50 | R@10 | R@50 |
| PALAVRA[5], ICCV'23 | 21.49 | 37.05 | 17.25 | 35.94 | 20.55 | 38.76 | 19.76 | 37.25 |
| SEARLE[2], ECCV'22 | 24.44 | 41.61 | 18.54 | 39.51 | 25.70 | 46.46 | 22.89 | 42.53 |
| CIReVL, ICLR'24 | 28.36 | 47.84 | 25.29 | 46.36 | 31.21 | 53.85 | 28.29 | 49.35 |
| CIReVL w/ $E^2$ (Ours) | **32.02** | **50.73** | **28.47** | **49.39** | **34.72** | **56.65** | **31.73** | **52.25** |

Figure 8: Results on Fashion-IQ dataset

| Model | Image to Text | | | Text to Image | | |
|---|---|---|---|---|---|---|
| | R@1 | R@5 | R@10 | R@1 | R@5 | R@10 |
| Full method | 62.8 | 89.3 | 95.3 | 64.5 | 90.1 | 95.5 |
| w/o Fusion Blocks | 58.8 | 88.7 | 94.9 | 58.6 | 88.9 | 94.9 |
| w/o Select. Tokens | 55.2 | 85.7 | 92.9 | 55.4 | 86.2 | 92.9 |
| w/o Region CL | 56.3 | 86.5 | 93.4 | 56.2 | 87.1 | 93.2 |
| w/ FILIP backed | 62.1 | 89.8 | 95.9 | 62.9 | 90.4 | 95.9 |

Figure 9: Ablations on FashionGen.

all of the image batch tokens and selection tokens. Without the explicit selection and fusion process in the fusion blocks, the selection tokens, which are supervised with region contrastive loss and carry rich information towards tag entities, are still interacting implicitly with image patch tokens and the global token in ViT layers. In this way, we assume that, even without fusion blocks, the global token still learns richer information about tag entities and lead to better retrieval performance. Then we are curious to see how much the explicit token selection and fusion process benefit the learning process, in addition to the potential improvements from this implicit interaction in transformer layers. Table 9 shows both modules are critical. We also ablate our region contrastive loss (RCL).

**Group of Selection Tokens.** We also study how each group of selection tokens contribute to the model performance. While all groups of selection tokens improve the model performance, their contributions are different. While Composition and Brand are more helpful, Season and Sub-category contribute slightly less. One potential reason is that they are easier to be visually distinguished than product texture and small logos, which especially require our fusion and selection process. We have detailed discussion on their effectiveness in Appendix F, and on parameter-efficiency in Appendix I.

**Parameter Sensitivity.** We also examine the impact of batch size, which has a significant influence on the performance of contrastive learning. $E^2$ consistency surpasses CLIP-FT across a range of batch sizes, with greater improvement over CLIP-FT as batch size decreases. We present detailed results and analysis in Appendix C.

**Number of Selection tokens.** In experiments, we use two selection tokens for each tag. It is reasonable as we are not selecting and fusing image patches in raw pixel space, instead, we conduct it with image patch (token) embeddings within ViT layers in contextual embedding space, where each token contains rich context/neighbor information. In fact, one token can already represent a large area if considering its neighbour information. Our choice of two tokens is empirically enough for all tags. We quantitatively validated this choice in Appendix I.

## 5 Limitations

Built upon the pre-trained foundation VL model (e.g., CLIP), $E^2$ could be bottlenecked by the qualify of its large-scale pre-training in general domain. Besides, as our regional contrastive learning with fusion blocks and selection tokens explicitly aligns regions in the input image with the corresponding tag entities from text, intuitively, the quality of the provided tag entities can influence our model performance. Although *in most cases*, fashion data contain such information (e.g., brand, composition etc.), it is still a limitation if only datasets with poor tag entities are available for finetuning.

## 6 Conclusion

In this paper, we propose a simple yet effective framework on learning fashion representations, and first emphasize the importance of learning fine-grained visual representations when applying CLIP to fashion domain. To achieve it, we further propose $E^2$ with selection tokens and region contrastive loss to enforce extra attention to details. Experimental results prove the effectiveness of $E^2$.

## Acknowledgement

The work is in part supported by the National Science Foundation under Grants IIS-2316306 and CNS-2330215, and a gift from Adobe.

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

## A    Experimental Settings

For FashionGen, following [27, 10], the model is trained for 20 epochs. The weight decay is set to $1e-4$, and the learning rate is set to $5e-5$ with the cosine annealing learning rate decay scheduler applied. As the selection tokens are randomly initialized, to match them with the pre-trained CLIP model, we first freeze other parameters and train selection tokens with an initial learning rate $5e-4$ for 5 epochs. The default batch size is 64. Configurations are the same with AmazonFashion, while the epoch is set to 10.

## B    Visualization

We provide more retrieval results in Fig. 15 and Fig. 16. For each example, the first row is the top-5 retrieval result by $E^2$ and the second row is the result by CLIP. Ground truth is boxed in green.

To intuitively see what is selected by selection tokens, we visualize selected tokens in the last stage in Fig. 14. Results show our selection tokens are effective in finding regions that contain detailed information about the tag entities, e.g., brand, composition. In the last transformer layer, only selection tokens and global token are kept (Eq. 4) (image tokens are dropped) to allow the latter effectively learns fine-grained representations from the former. We also visualize what the global token of CLIP focuses by marking image tokens with the maximum attention values with it for each attention head (12 in total). Note the overlap exists, i.e., one image token may have the maximum attention value with the global token in multiple attention heads. Results show CLIP is ineffective in finding tag entity-related information and sometimes focuses on totally unrelated regions. It is observed that visually dominant areas which are closer to the global view of the product, e.g., dark areas in a dark product in Fig. 14 (g), (i), (j), tend to win higher attention scores with the global token. Consequently, regions with detailed information such as logos are overlooked. One potential reason is that, pre-training data on general domain often consist of a simple caption and a few very distinctive objects (Fig. 2), which enables CLIP to capture features better from a high-level view. In this case, focusing more on a global view of objects instead of a local view helps the model to differentiate large and distinctive object in an image. However, when it comes to fashion domain, where multiple details of a single product are required to be aware of, CLIP fails because CLIP still tries to capture the global view of an object and ignores local view, even through it has already been finetuned. By contrast, with our model, *Brands* are all picked. *Composition* prefers solid color areas, where material is more clear. *Sub-category* is reflected by sleeves, collar and shoulder. Sleeves and front zippers reveal the *Season*.

## C    Parameter Sensitivity

We also investigate the impact of batch size, which can be substantial to contrastive learning. Results in Tab. 5 show that $E^2$ consistency outperforms CLIP-FT across different batch sizes. Our findings indicate that $E^2$ consistency surpasses CLIP-FT across a range of batch sizes, with $E^2$ showing greater improvement over CLIP-FT as batch size decreases. Notably, it is especially helpful in scenarios where hardware limitations impose constraints on batch size, where $E^2$'s superiority becomes more evident.

| Batch Size | Model | Image to Text | | | Text to Image | | | SumR |
|---|---|---|---|---|---|---|---|---|
| | | R@1 | R@5 | R@10 | R@1 | R@5 | R@10 | |
| 16 | CLIP-FT | 29.9 | 63.0 | 76.7 | 30.7 | 63.7 | 76.7 | 340.7 |
| | $E^2$ (Ours) | **42.0** | **75.7** | **86.3** | **42.2** | **76.5** | **87.0** | **409.7** |
| 32 | CLIP-FT | 42.7 | 77.0 | 87.0 | 43.6 | 77.1 | 87.1 | 414.6 |
| | $E^2$ (Ours) | **53.3** | **84.4** | **92.7** | **53.9** | **85.3** | **92.9** | **462.6** |
| 64 | CLIP-FT | 55.2 | 85.7 | 92.9 | 55.4 | 86.2 | 92.9 | 468.2 |
| | $E^2$ (Ours) | **62.8** | **89.7** | **95.3** | **64.5** | **90.1** | **95.5** | **497.9** |
| 128 | CLIP-FT | 62.5 | 88.2 | 93.7 | 62.7 | 88.7 | 93.9 | 489.7 |
| | $E^2$ (Ours) | **66.9** | **91.3** | **95.9** | **66.6** | **91.3** | **95.8** | **507.7** |

Table 5: Influence of batch size. We evaluate the performance of $E^2$ and CLIP-FT under different batch size settings on FashionGen.

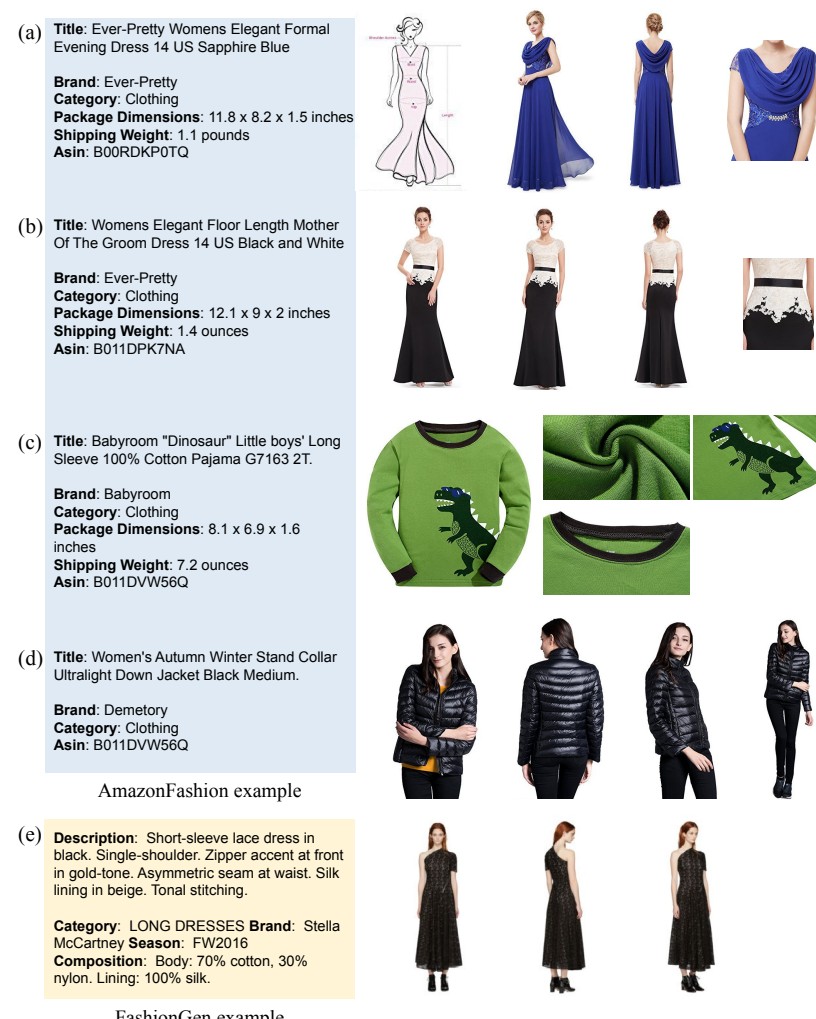

(a) **Title**: Ever-Pretty Womens Elegant Formal Evening Dress 14 US Sapphire Blue

**Brand**: Ever-Pretty
**Category**: Clothing
**Package Dimensions**: 11.8 x 8.2 x 1.5 inches
**Shipping Weight**: 1.1 pounds
**Asin**: B00RDKP0TQ

(b) **Title**: Womens Elegant Floor Length Mother Of The Groom Dress 14 US Black and White

**Brand**: Ever-Pretty
**Category**: Clothing
**Package Dimensions**: 12.1 x 9 x 2 inches
**Shipping Weight**: 1.4 ounces
**Asin**: B011DPK7NA

(c) **Title**: Babyroom "Dinosaur" Little boys' Long Sleeve 100% Cotton Pajama G7163 2T.

**Brand**: Babyroom
**Category**: Clothing
**Package Dimensions**: 8.1 x 6.9 x 1.6 inches
**Shipping Weight**: 7.2 ounces
**Asin**: B011DVW56Q

(d) **Title**: Women's Autumn Winter Stand Collar Ultralight Down Jacket Black Medium.

**Brand**: Demetory
**Category**: Clothing
**Asin**: B011DVW56Q

AmazonFashion example

(e) **Description**: Short-sleeve lace dress in black. Single-shoulder. Zipper accent at front in gold-tone. Asymmetric seam at waist. Silk lining in beige. Tonal stitching.

**Category**: LONG DRESSES **Brand**: Stella McCartney **Season**: FW2016
**Composition**: Body: 70% cotton, 30% nylon. Lining: 100% silk.

FashionGen example

Figure 10: Examples from AmazonFashion (blue) and FashionGen (yellow).

# D Datasets

FashionGen [36] is the existing benchmark dataset for fashion cross-domain retrieval. Existing works [12, 10, 27, 56] evaluate the performance of their model on fashion cross-domain retrieval with this dataset.

FashionViL [12] uses two dataset: FashionGen [36] and FashionIQ [47] for fashion domain retrieval. The critical difference is: FashionIQ [47] is for text-guided image retrieval (TGIR), a special type of image retrieval problem, while FashionGen [36] is for cross-modal retrieval. Similar to FashionGen, our AmazonFashion is built for the task of cross-modal retrieval.

Compared with M5Product [7] and Product1M [51], ours has two distinctive advantages: (a) Larger *fashion* data size: Product1M is designed for *cosmetics* and *groceries*, not for *fashion products*. In all 6.3M data in M5Product, 0.5M is in *fashion* domain, while ours is 1.3M. (b) 3.1 times conciser caption in average, leading to a more practical case: in practice, users usually search products online with short queries, instead of a long paragraph of detailed text in FashionGen style. Our dataset better matches this more challenging and practical case.

| Dataset | Image-Text Pairs | Brand Count | Product Count | Avg. Desc. Len. |
|---|---|---|---|---|
| FashionGen | 29.6K | 570 | 67666 | 28.92 |
| AmazonFashion (Ours) | 1.3M | 38211 | 544713 | 11.43 |

Table 6: Data statistics of AmazonFashion and FashionGen. We show the number of image-text pairs, brand count, product count and the average length of the image descriptions.

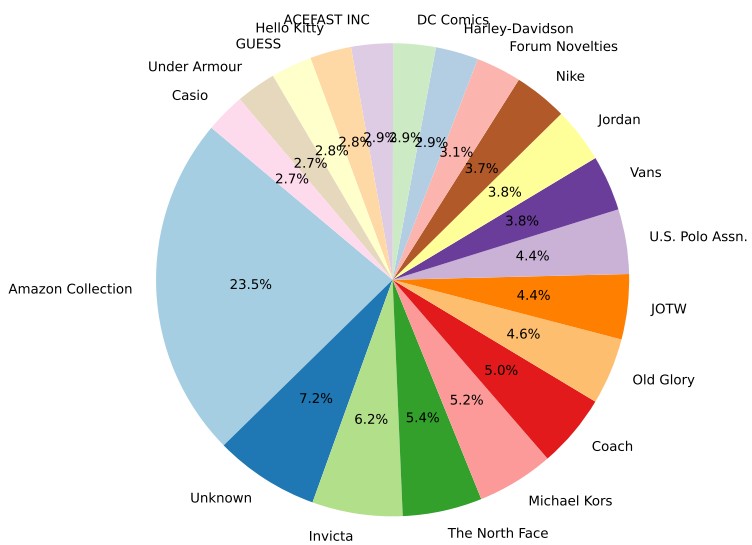

Figure 11: Top 20 most frequent brands in FashionGen.

## D.1 FashionGen

It contains $67,666$ different fashion products. Each product has one text description with one to six images from different angles. There are $260,480$ and $35,528$ image-text pairs for training and testing,

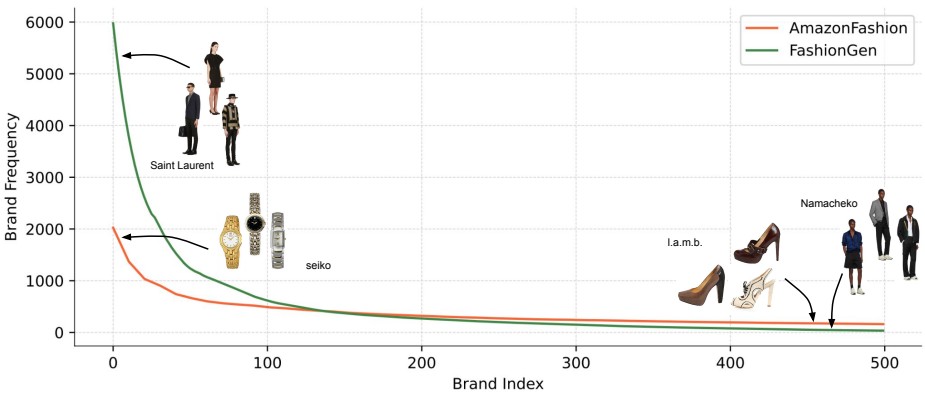

Figure 12: Brand frequency of AmazonFashion and FashionGen.

respectively. Each text description contains an overall description along with several tag entities: Brand, Sub-category, Season and Composition.

## D.2 AmazonFashion

We show several examples from AmazonFashion and FashionGen in Fig. 10 to highlight our differences between these two benchmarks. We also present detailed statistics of two datasets in Tab. 6 and Fig .12. In general, AmazonFashion is closer to the practical cross-model retrieval scenario.

Tab. 6 shows that AmazonFashion is different from FashionGen in notably larger size (1.3M), a wider variety of brands and products (67 and 8 times more, respectively), more concise and general descriptions (0.6 times shorter), and more diverse image scopes. These differences make AmazonFashion both more challenging and closer to real-world scenarios, where query languages from users are often more general and more concise, meanwhile, the number of candidate products is usually notably greater.

We also present the brand frequency plot in Fig .12. For each dataset, we selected the top 200 brands with the highest frequency of occurrence and plotted them in a descending order. Note that for the AmazonFashion plot, we excluded the brand *amazon collection*, which has a frequency of $19, 714$, significantly higher than the other brands. Both datasets exhibit a long-tail distribution in terms of brand frequency, which accurately reflects real-world scenarios.

Besides significant differences in data statistics, images in AmazonFashion also exhibit substantial variations. While FashionGen only provides regular views of a product from different angles such as Fig. 10(e), AmazonFshion includes more diverse images, such as product pictures from different domains in Fig. 10(a), and high-resolution images that showcase intricate product details in Fig. 10(a-c). It is close to practical scenarios where available product images on website often vary in styles. It also brings new challenges as better adaption to images at different scales and styles is excepted. Thus, the AmazonFashion holds significant value for researches on cross-modal retrieval in the fashion domain, and we encourage its use in future work.

## D.3 Model Results

We report results of representative CLIP-based models, including ours on AmazonFashion in Tab. 7.

## D.4 Brand Details

We show the distribution of the frequency of top 20 most frequent brand in Fig 11.

| Model | Image to Text | | | Text to Image | | | *SumR* |
|---|---|---|---|---|---|---|---|
| | R@1 | R@5 | R@10 | R@1 | R@5 | R@10 | |
| FILIP-FT* [49] | 6.2 | 17.8 | 25.8 | 6.2 | 18.1 | 25.9 | 100.0 |
| CLIP-FT [34] | 6.1 | 17.7 | 25.8 | 6.2 | 17.9 | 25.8 | 99.5 |
| $E^2$ (Ours) | **7.5** | **20.3** | **28.4** | **7.4** | **20.2** | **28.4** | **112.2** |

Table 7: Retrieval performances (sample-100 evaluation) on AmazonFashion. (*) denotes results from our implementation.

# E   Linear Probing

## E.1   Experiment

Linear probing is a popular technique to evaluate the quality of learned embeddings of a model: fitting a linear classifier to the image embeddings of the model. Typically, the better the model is trained, the higher performance a classifier achieves, because a well-trained model tends to give more informative embeddings, from which a classifier learns better.

To evaluate how well CLIP, CLIP-FT, FILIP-FT and $E^2$ are able to learn entity-specific knowledge from images, we perform four classification tasks. More specifically, for each task, we train a linear classifier towards one of four tag entities (i.e., to classify: Brand, Sub-category, Season and Composition) on image embeddings of CLIP [34], CLIP-FT [34], FILIP-FT [49], and $E^2$, respectively. We use the Adam optimizer and train each classifier for 20 epochs with learning rate = 0.001. Results are shown in Fig. 3.

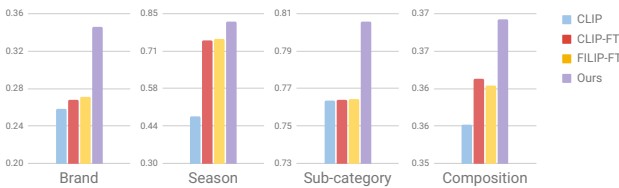

Figure 13: Linear probing results on FashionGen. The more informative embeddings are, the higher accuracy a classifier obtains.

## E.2   Analysis

Results show classifiers trained with $E^2$ embeddings consistently outperform classifiers trained with CLIP, CLIP-FT and FILIP-FT embeddings, indicating that $E^2$ embeddings contain more information about tag entities. This can be attributed to our proposed token fusion, token selection, and region contrastive loss, which explicitly forces the model to pay special attention to these details.

Besides, we delve deeper into the linear probing results, which further reveal several interesting facts on how each tag entity impacts the model performance. Fig. 13 (Brand) indicates that while fine-tuning can enhance CLIP's understanding of product brands, the extent of improvement is comparatively small when contrasted with the significant progress achieved through $E^2$. By comparison, Fig. 13 (Season) shows fine-tuning is effective for CLIP to learn about produce season. However, when it comes to Fig. 13 (Sub-category), fine-tuning almost yields no benefits. We assume that fine-tuning can only help learn easy classes such as bag, jeans, shirts, etc., while $E^2$ can help difficult classes. In this case, because the vanilla pre-trained CLIP is already effective in recognizing high-level concepts, fine-tuning fails to yield further improvements. $E^2$ significantly benefits the learning of product categories by learning more about difficult classes. In Fig. 13 (Composition), although the relative magnitudes are similar to that in Fig. 13 (Brand), the absolute differences among four models are much smaller, which indicates effectively learning about product composition is a harder task.

# F    Ablation

To study how each group of selection tokens contribute to the model performance, we train and evaluate $E^2$ without one group of selection tokens associated with one tag entity at a time and report the result in Table 8. While all groups of selection tokens improve the model performance, their contributions are different. Selection tokens associated with Brand and Composition are more helpful. As Fig. 15 and Fig. 16 show, paying special attention to the brand and composition of a product helps our model better differentiate visually very similar items. By contrast, Season and Sub-category contribute less as they are easier to be visually distinguished than product texture and logos, which are harder to discriminate and especially requires our fusion and selection process with the selection tokens.

| Model | Image to Text | | | Text to Image | | | SumR |
|---|---|---|---|---|---|---|---|
| | R@1 | R@5 | R@10 | R@1 | R@5 | R@10 | |
| $E^2$ (Ours) | 62.8 | 89.7 | 95.3 | 64.5 | 90.1 | 95.5 | 497.9 |
| w/o Composition | 59.4 | 88.4 | 94.2 | 61.8 | 89.9 | 95.3 | 488.9 |
| w/o Sub-category | 61.2 | 89.7 | 95.0 | 60.8 | 89.6 | 95.1 | 491.3 |
| w/o Brand | 58.3 | 88.3 | 95.0 | 58.0 | 88.5 | 94.7 | 482.7 |
| w/o Season | 60.0 | 88.9 | 94.4 | 59.8 | 88.6 | 95.0 | 486.6 |

Table 8: Ablation study (selection tokens) on FashionGen.

# G    Innovation

In this section, we discuss our differences from related works, and our innovation (motivation and technical contribution).

## G.1    Technical Innovation

The design of selection and fusion mechanism with fusion blocks and selection tokens is our new technical contribution, which notably improves the performance. Typical CLIP-based adaption models (e.g., **RegionCLIP** [54], **FILIP** [49], DenseCLIP [35], **FashionCLIP** [4]) are built upon CLIP without modifying its inner structure. However, we break into the CLIP encoder by introducing tag-aware fusion blocks. Specifically, the above models, including **FashionSAP** [14], *do not explicitly align image patch tokens within ViT layers with text information*, which we prove to be critical in fashion domain where the details do matter. To this end, our tag-aware selection tokens select and fuse image patch tokens. They differ from typical new additional tokens (VPT [18], DeiT [43]), which are simply added to input layers and structurally similar to the *CLS* token. Our adaptation of CLIP is new and first tailored for fashion scenarios with tag entities, where general CLIP-adaptation models are hard to handle (more details in Appendix H).

## G.2    Motivation

Our highlight is not improving the CLIP vision encoder in general domain, as existing works such as FILIP [49] do. *Instead, we are among the first to adapt CLIP to fashion domain.* When combined with SOTA CLIP-based zero-shot image captioning models, their performance can be obviously improved.

Most recently, several works [49, 21, 11] start to improve language image pre-training with semantic alignment. FILIP [49] and LOUPE [21] learn better representations with semantic alignment of patch tokens. HiCLIP [11] learns to discover hierarchy in data. GroupViT [48] also forms semantic segments hierarchically, but it only focuses on semantic segmentation. Our motivation differs from above methods in that, while they learn representations of larger objects via grouping smaller regions (i.e., image tokens), we focus on certain regions with detailed information of an object, e.g., its brand and composition. Besides, they focus on designing models for better image-text matching generally without considering the uniqueness of fashion data. Our $E^2$ pays additional attention to image details associated with *tag entities* for better visual learning (more details in Appendix H).

| Model | Image to Text | | | Text to Image | | | *SumR* |
|---|---|---|---|---|---|---|---|
| | R@1 | R@5 | R@10 | R@1 | R@5 | R@10 | |
| EI-CLIP [27] | 59.4 | 88.4 | 94.2 | 61.8 | 89.9 | 95.3 | 488.9 |
| EI-CLIP [27] w/o E. | 56.2 | 86.8 | 93.6 | 56.4 | 87.2 | 93.5 | 473.7 |
| $E^2$ (Ours) | **62.8** | **89.7** | **95.3** | **64.5** | **90.1** | **95.5** | **497.9** |

Table 9: Retrieval performances (full-candidate) on FashionGen.

## H    Discussion

**Differences from Segmentation Models.** Some semantic segmentation models [39, 48] also involve additional tokens, including GroupViT [48]. We are very different from each other in: (1) motivation, (2) selection technical details, and (3) overall framework. In summary, group tokens in GroupViT serve as "collectors", thus **image tokens** are iterated to be allocated to a group token. By contrast, in $E^2$, the process is opposite: **selection tokens** are iterated to select one most relevant image token. We do attention with different objects therefore are not interchangeable. GroupViT cannot select the most relevant image tokens, while ours does not support grouping of them. Our framework design is also significantly different. GroupViT adopts an iterative approach, where grouped tokens are consecutively fed as inputs to transformer layers, introducing new group tokens at each stage. In contrast, our approach utilizes image tokens along with a fixed number of selection tokens as inputs. To sum up, due to different tasks/objectives, GroupViT and $E^2$ are very different in both technical details and the overall framework.

**Comparisons with EI-CLIP.** While we facilitate the image encoder, EI-CLIP [27] improves the text encoding process by reducing word ambiguity in tag entities with additional text encoders. Thus we are orthogonal to each other. Besides, EI-CLIP uses additional text encoders and during inference, tag entities must be separately encoded by the encoders. In contrast, $E^2$ does not require separated tag entities as text input. In our model, same to existing works [10, 56] and CLIP, tag entities are concatenated with the description as one string, which is input to the CLIP text encoder. This is more practical and especially helpful in cases where tag entities are mixed with descriptions and separating them is infeasible. We also evaluate EI-CLIP under this setting, which follows existing works [10, 56] and ours (w/o E. in Table 9). Although EI-CLIP has additional encoders and requires separated tag entities during inference, it is still inferior to ours according to Table 9. When evaluating under the same setting, the gap is more notable.

**Differences from Fine-grained CLIPs.** Several recent works [49, 11, 21] propose to better match image-text pairs with semantic alignment or hierarchical grouping. While they learn representations of larger objects via grouping smaller regions (image tokens), we focus on finding certain regions with detailed information of part of an object (e.g., its brand, season, composition), motivated by the uniqueness of fashion data. We achieve this by designing a better way to exploit *tag entities*, i.e., introducing selection tokens associated with these entities, while they seek for a general way to better align an image-text pair without considering the of characteristics of fashion data. Thus our work is orthogonal to theirs and we can build $E^2$ based on their pre-trained models. We choose CLIP in this work as it is the most representative one, based on which makes it extendable to others. Because codes and pre-trained models of mentioned CLIP variants are not released, we are not able to evaluate all of them. However, as FILIP [49] shares the same structure with CLIP, we implement it and initialize it with a pre-trained CLIP model and finetune it on our datasets. We also use FILIP as the backbone model in our ablation study.

## I    Analytical Experiments

In this section, we explore 1. how the number of selection tokens can influence our model performance and 2. the parameter efficiency of our model.

## I.1 Selection Tokens

In Tab. 10, we provide different choices of selection tokens (②, ④-⑧). Two tokens for each tag are better than one (④), and comparable to more tokens (⑤-⑦), or a combination of different numbers (⑧). It is reasonable as we are not selecting and fusing image patches in raw pixel space, instead, we conduct it with image patch (token) embeddings within ViT layers in contextual embedding space, where each token contains rich context/neighbor information. E.g., in Fig. 7 in paper, one token can already represent a large area if considering its neighbour information. Our choice of two tokens is empirically enough for all tags, including *season*, as quantitatively validated in Tab. 10.

## I.2 Parameter Efficiency

We compare the amount of learnable parameters of ours (default setting in paper is shaded in green) and most competing baselines (⑨ - ⑫) in Tab. 10. Compared to CLIP-FT (⑩), we only involve minimal new parameters in fusion blocks: (1) three projection matrices: $W^v$, $W^k$ and $W^q$ $\in \mathbb{R}^{768 \times d_W}$, and (2): 8 selection tokens $s \in \mathbb{R}^{1 \times 768}$, where $d_W = 768$ in paper. We have comparable parameters with CLIP-based methods and significantly fewer parameters than ⑫, while achieving notably better results over them. We also tried different $d_W$ and numbers of selection tokens (①-⑧), and our original setting (②) achieves competing results with very few new parameters.

| Model | C | S | B | SC | Params (*Total*) | Params (*Token*) | SumR |
|---|---|---|---|---|---|---|---|
| ① Ours ($d_W$=128) | 2 | 2 | 2 | 2 | 86.6M | 6.1K | 494.3 |
| ② Ours ($d_W$=768) | 2 | 2 | 2 | 2 | 87.7M | 6.1K | 497.9 |
| ③ Ours ($d_W$=2048) | 2 | 2 | 2 | 2 | 94.8M | 6.1K | 498.2 |
| ④ Ours ($d_W$=768) | 1 | 1 | 1 | 1 | 87.7M | 3.1K | 493.8 |
| ⑤ Ours ($d_W$=768) | 4 | 4 | 4 | 4 | 87.7M | 12K | 497.9 |
| ⑥ Ours ($d_W$=768) | 8 | 8 | 8 | 8 | 87.7M | 24K | 498.3 |
| ⑦ Ours ($d_W$=768) | 16 | 16 | 16 | 16 | 87.7M | 48K | 498.2 |
| ⑧ Ours ($d_W$=768) | 4 | 8 | 4 | 8 | 87.7M | 18K | 497.9 |
| ⑨ EI-CLIP | - | - | - | - | 89.1M | - | 473.7 |
| ⑩ CLIP-FT | - | - | - | - | 86M | - | 468.2 |
| ⑪ FILIP-FT | - | - | - | - | 86M | - | 470.2 |
| ⑫ FashionSAP | - | - | - | - | 120M | - | 451.8 |

Table 10: Results on FashionGen with *full-candidate* evaluation. **C**: Composition, **S**: Season, **B**: Brand, **SC**: Sub-category. Numbers are the quantity of corresponding selection tokens. Total number of learnable parameters in vision encoder: **Params** (*Total*). We also list parameters of selection tokens (part of the *total* **Params**): **Params** (*Token*).

# J  Broader Impacts

There is no special societal impact of our work performed.

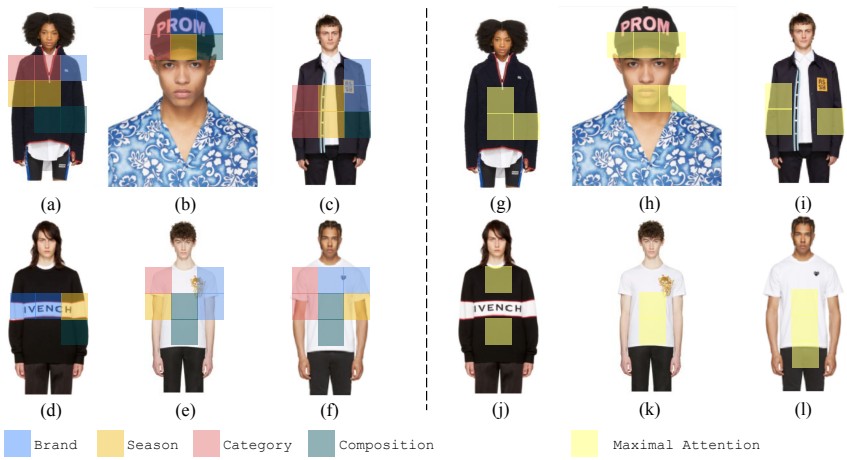

Figure 14: **Left:** $E^2$ **Token Selection Examples.** We marked selected image patch tokens during the second stage. Each tag entity is associated with two selection tokens. *Note that tokens may overlap when an image patch token is selected by multiple selection tokens.* **Right: CLIP Attention Visualization.** We marked image tokens with the maximum attention values (in each attention head) with the global token. ***Note that, the "Category" in the figure refers to "Sub-category".***

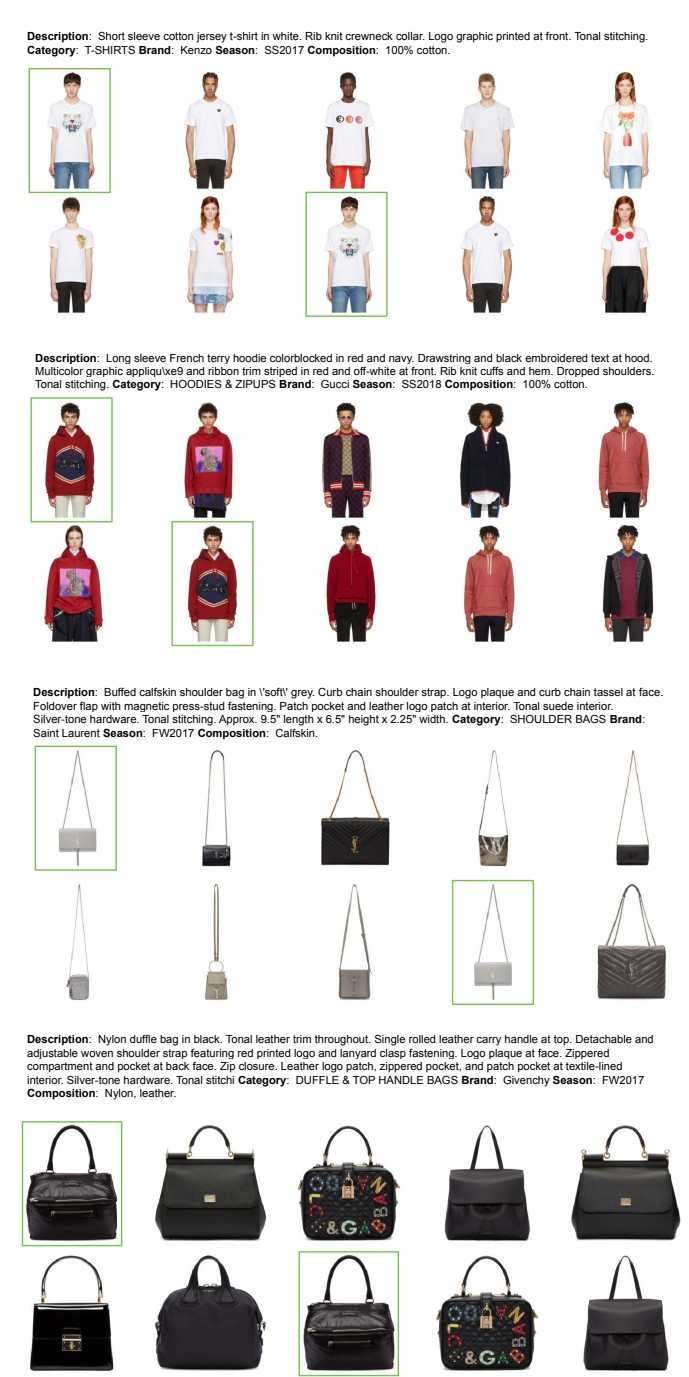

Figure 15: **T2I retrieval examples.** For each example, the query text is displayed on the top. The first row is the top-5 retrieval result by $E^2$ and the second row is the result by CLIP. Ground truth is boxed in green.

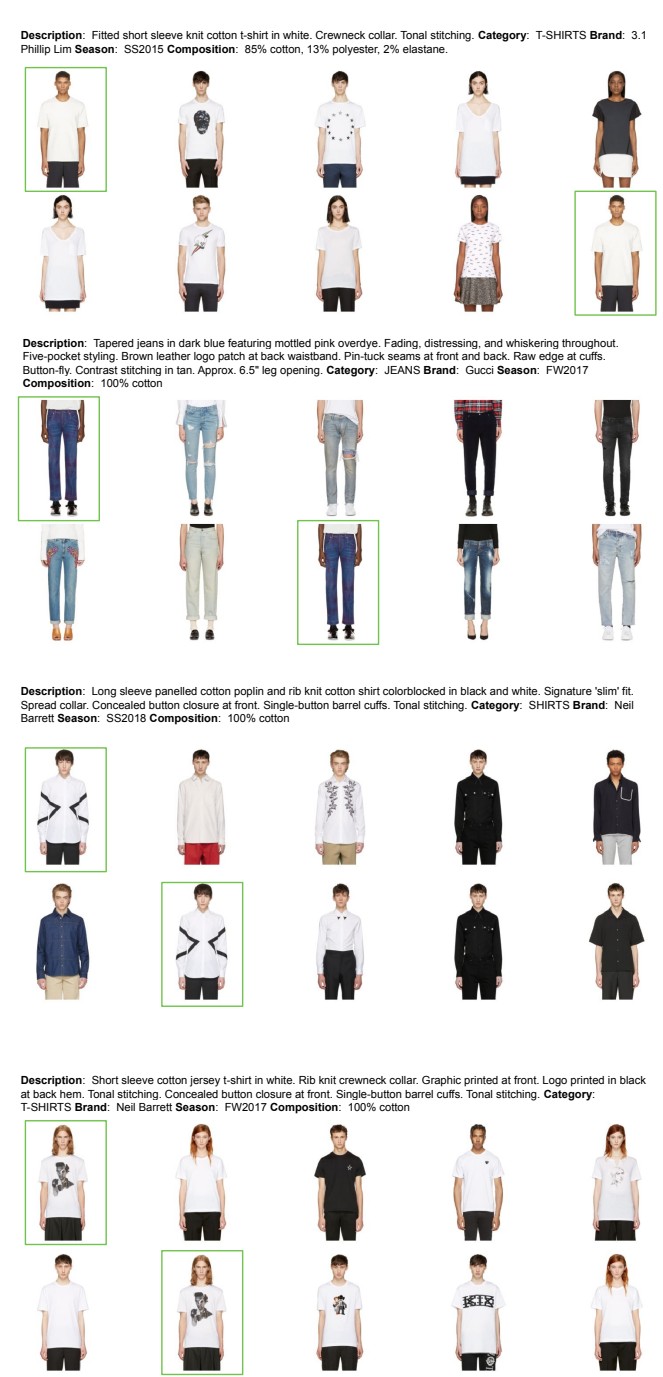

Figure 16: **T2I retrieval examples.** For each example, the query text is displayed on the top. The first row is the top-5 retrieval result by $E^2$ and the second row is the result by CLIP. Ground truth is boxed in green.

