# OpenReview forum: "Easy Regional Contrastive Learning of Expressive Fashion Representations"
_NeurIPS.cc/2024/Conference — NeurIPS 2024 poster_

### Official Review · Reviewer_LcUx · 2024-07-07

**Soundness:** 4
**Presentation:** 4
**Contribution:** 3
**Rating:** 6
**Confidence:** 4

**Summary:**

This paper focused on adapting CLIP-based VLMs to the fashion domain. The motivation is that directly finetuning CLIP models on fashion data will lead to insufficient learning of entity-related details like logos and composition. To tackle this challenge, this work proposed a region-based contrastive loss by inducing selection tokens to capture information on tag entities like brand, sub-category, and season, and further applied region contrastive loss on these token embeddings. Also, this work collected a new dataset called AmazonFashion which consists of briefer and more general text descriptions compared to the previous FashionGen benchmark. Experiments on several benchmarks validated the effectiveness of the proposed method.

**Strengths:**

\+ The contributions of this work to the fashion domain are significant in terms of method and dataset.

First, this work introduced selection tokens of tag entities and region contrastive loss to adapt CLIP models to the fashion domain and learn more fine-grained visual representations. The motivation that directly finetuning CLIP models on fashion data will lead to insufficient learning of entity-related details is reasonable and the solution makes sense.

Second, this paper collected a new dataset called AmazonFashion which consists of briefer and more general text descriptions compared to the previous FashionGen benchmark. The new dataset would serve as a new benchmark for future research on the fashion domain.

\+ The paper is well-organized and written. It is easy to follow the idea and understand the key contributions of the work.

\+ Experiments are conducted on Fashion-Gen and AmazonFashion datasets and evaluated on cross-model retrieval, zero-shot image captioning, and zero-shot text-guided image retrieval tasks to validate the effectiveness of the proposed method. The method can be applied to various base models like CapDec and DeCap, which shows the generalization of the method.

**Weaknesses:**

\- The proposed method works well for the fashion domain but is not evaluated on general domains for other fine-grained recognition or captioning tasks. This is not a reason to reject the paper, but limits the scope of the work. This work is a good application work and proposed a reasonable solution, but the insights are not very strong for other domains or tasks.

**Questions:**

I have no further questions for the authors.

**Limitations:**

The authors have adequately addressed the limitations.

---

> ### Author Rebuttal · Authors · 2024-08-03
>
> We sincerely thank you for your efforts in reviewing this paper and giving valuable suggestions!
>
> We appreciate your recognition of the effectiveness and rationale of our work. While our method consistently outperforms existing works and exhibits simplicity and efficiency in fashion domain, it also demonstrates some insights in finetuning large-scale pre-trained vision-language models on paired (image-text) data with additional attributes.
>
> Our design could be viewed as a general vision-language contrastive learning method for paired image-text data with attributes. As our model learns to attend to important attributes/regions with additional fusion blocks and selection tokens, our method could be applicable to other data/domains as long as specific attributes are provided.
>
> In fashion domain, the specific attributes are the existing tag entities, in other domains (such as medical images), we could directly use similar existing attributes (if any) and apply our model.
>
> If there are no such attributes in existing data immediately,  one intuitive solution could be collecting useful attributes for existing data first (manually or using other models).  For example, in contrastive learning for medical images [2,3,4,5], extracted entities [6] from image descriptions could be used as additional attributes.
>
> Finally, we would like to kindly remind that, fashion domain itself is an emerging topic [7-15], and our effective contrastive learning of fashion representations would benefit a series of downstream tasks such as cross-modal retrieval, fashion captioning, text-guided retrieval, recommendation, etc.
>
> Although our method also demonstrates insights in finetuning large-scale pre-trained vision-language models on any paired (image-text) data with additional attributes, and could be potentially applied to other domains (such as medical images), the primary focus of this work is still fashion representation learning [8,10,12,13,14,15].
>
> Thank you again for your thorough review of this paper and your valuable suggestions. We hope the response could address your concerns. If you have any further questions, we look forward to more discussion with you!
>
> # Reference
>
> (available in "Author Rebuttal by Authors")

---

### Official Review · Reviewer_Kwtu · 2024-07-09

**Soundness:** 3
**Presentation:** 2
**Contribution:** 3
**Rating:** 6
**Confidence:** 4

**Summary:**

This paper proposes a framework to train CLIP models more adaptable to fashion domain (shopping item images and structured descriptions, i.e, “tags”, such as brand, composition etc.). The new framework (“E2”) outperforms other common methods by a large margin on a wide range of tasks, by 1) adding fusion blocks in image encoder to select more representative patches for tags, and 2) adding tag specific contrastive losses.

**Strengths:**

The proposed method is simple yet effective. It demonstrates the effectiveness of the region level visual patch selection and fusion, which also makes sense for applications beyond the fashion domain.

The performance improvement is large and consistent on diverse VL tasks.

It’s easy to replace CLIP components with E2 in any VLM. This paper also shows its effectiveness beyond retrieval by replacing the CLIP in DeCap.

**Weaknesses:**

The efficiency evaluation is missing. It’s unclear how the training/inference speed is affected by the newly added components in CLIP.

Besides retrieval and captioning, visual question answering is another common task for vision language modeling. It would be more solid if we add another eval on CLIP based VQA models.

The impact is mostly limited in the fashion domain. It would be more impactful if more applications would be found for the proposed method.

**Questions:**

It’s common to have the same tags from different images (such as the same brand of different images of t-shirts, shoes…). This could lead to many false-negatives in contrastive training, and may explain why the fusion block is necessary. Have you identified such an issue and considered any solution for this (such as “soft” contrastive labels)?

In Figure 5, it would be more clear if the locations of a/b/c can be explicitly stated, such as “a - Left, b - Right, c - Middle”.

What is the token length of text encoder?

In  Parameter Sensitivity section, is the batch size’s unit thousand (otherwise looks too small for contrastive learning)?

---

> ### Author Rebuttal · Authors · 2024-08-03
>
> We sincerely thank you for your efforts in reviewing this paper and giving valuable suggestions!
>
> ## W1
>
> Thanks for your helpful suggestion! One of the highlights of our model is being lightweight with minimal additional cost.
>
> We compare the training and inference speed with the vanilla CLIP as follows:
>
> Under the same environment and configuration, with a single A100 (40GB), the running time is summarized in the table:
>
>
>
> | Model        | Training Time (per epoch) | Inference Time |
> | ------------ | ------------------------- | -------------- |
> | CLIP         | 231.86s (3 m 52 s)        | 17.55s         |
> | $E^2$ (Ours) | 250.60s (4m 10 s)         | 17.57s         |
>
> We report the averaged training time of one epoch on FashionGen (20 epochs in total for full finetuing). For inference, we report the running time for single inference with full-candidate evaluation on FashionGen (390K candidate samples).
>
> The above table shows that we have only around 8% additional training time (19 seconds longer per epoch) compared to CLIP, while the inference time is almost the same.
>
> ## W2
>
> Thanks for the very good suggestion, and an interesting direction as future work. We currently do not have the evaluation on VQA because, although there are a lot of available well-established VQA benchmarks in the general domain, there is not a standard VQA benchmark or dataset in the fashion domain. FashionVQA [16] is the most related work, unfortunately neither datasets nor models are publicly available.
>
> Therefore, if we want to evaluate CLIP-based VQA models (and our model) in the fashion domain, we need to at least build a fashion VQA benchmark first (and possibly another fashion dataset for simple finetuning), which is beyond the scope of this work.
>
> Because our primary focus is not benchmarking fashion VQA, we currently do not evaluate VQA. We mainly focus on the well-defined and well-established fashion tasks in existing works (such as ALBEF, SyncMask, FashionSAP, FashionViL, FashionBert, KaleidoBert, FaD-VLP, and FAME-ViL).
>
> ## W3
>
> Thanks for the great advice. As our model learns to attend to important attributes/regions with additional fusion blocks and selection tokens, our method could be easily applicable to other domains as long as specific attributes are provided (like tag entities in fashion domain).
>
> In fashion domain, the specific attributes are the existing tag entities, in other domains (such as medical images), we could directly use similar existing attributes (if any) and apply our model.
>
> If there are no such attributes in existing data immediately,  one intuitive solution could be collecting useful attributes for existing data first (manually or using other models).  For example, in contrastive learning for medical images [2,3,4,5], extracted entities [6] from image descriptions could be used as additional attributes.
>
> Finally, we would like to kindly remind that, fashion domain itself is an emerging topic [7-15], and our effective contrastive learning of fashion representations would benefit a series of downstream tasks such as cross-modal retrieval, fashion captioning, text-guided retrieval, recommendation, etc.
> Although our method also demonstrates insights in finetuning large-scale pre-trained vision-language models on any paired (image-text) data with additional attributes, and could be potentially applied to other domains (such as medical images), the primary focus of this work is still fashion representation learning [8,10,12,13,14,15].
>
> ## Q1 & Q4
>
> Thanks for this insightful question. For question 4, the batch size's unit is not thousand. It is understandable that in large-scale pre-training in general domain (e.g., 400M samples in CLIP/OpenCLIP[17] pre-training), large batch sizes (16K, 32K, or 88K) are used.
> Such large batch sizes are the “global batch size”, which is accumulated from local batch sizes in thousands of individual GPUs. For instance, OpenCLIP [17] maximizes the local batch size per GPU (86 to 88 per GPU) and uses around one thousand GPUs to have a global batch size of 86K to 88K.
> When adapting the pre-trained CLIP to a specific domain (FashionGen) with obviously fewer data and limited GPU resources, a batch size from 16 to 128 is typically used in existing works (FashionViL, FashionSAP, FaD-VLP, and FAME-ViL). We follow them to set similar batch sizes for fair comparison.
>
> This could also explain the question 1. We understand that there might be false-negatives in contrastive training in a batch, because different images could have the same tag. However, as the commonly used batch size for the in-domain finetuning on fashion data is relatively small (16 to 128), in practice, false-negatives are not common in a single batch. Therefore, we currently just ignore their marginal influence and do not include a specific design for it.
>
> However, we agree that involving a specific solution for this would be a bonus. Potential solutions could include adopting different sampling strategies or adjusted objectives  (e.g., Khosla et al. [18] propose “supervised contrastive” (fig. 2 in their paper) to contrast the set of all samples from the same class as positives against the negatives from the rest of the batch)..
>
> ## Q2
>
> Thanks for your advice. It is very helpful for improving the clarity of this figure.
>
> ## Q3
>
> The default limit for the input text length of the CLIP text encoder we used is 77
>
> Thank you again for your thorough review of this paper and your valuable suggestions. We hope the response could address your concerns. If you have any further questions, we look forward to more discussions with you!
>
> # Reference
>
> (available in "Author Rebuttal by Authors")

---

> > ### Comment · Reviewer_Kwtu · 2024-08-10
> >
> > Thanks for the detailed explanation, which has addressed most of my questions. I am happy to maintain my previous rating.

---

> > > ### Author Response · Authors · 2024-08-11
> > >
> > > Thank you for your prompt response! We are glad to hear that our explanation addressed most of your questions and that you are comfortable maintaining the positive rating. Your feedback is greatly appreciated!

---

### Official Review · Reviewer_tu2V · 2024-07-17

**Soundness:** 2
**Presentation:** 2
**Contribution:** 2
**Rating:** 5
**Confidence:** 3

**Summary:**

The paper discusses adaptation of CLIP for fashion domain w/ an emphasis on importance of learning fine-grained visual representations. The authors propose E^2 with selection tokens and region contrastive loss to enforce extra attention.

**Strengths:**

Emphasis on importance of learning fine-grained visual representations with selection tokens and region contrastive loss to enforce extra attention. Good analysis is performed in the experiments.

**Weaknesses:**

Minor comment: the paper could be made a bit easier to understand for the general ML audience attending NeurIPS

**Questions:**

none

**Limitations:**

not mentioned

---

> ### Author Rebuttal · Authors · 2024-08-03
>
> Thank you for your  valuable suggestions and the recognition of the analysis and experiments in our study!
>
> In our final version, we will make the paper more understandable for the general ML audience in NeurIPS. Specifically, we will carefully introduce the background of our problem, and how our specific approach is derived from general machine learning principles.

---

### Official Review · Reviewer_Mfk5 · 2024-07-22

**Soundness:** 3
**Presentation:** 3
**Contribution:** 3
**Rating:** 5
**Confidence:** 5

**Summary:**

This work proposes a new approach to learn improved and more expressive visual representations for fashion-specific tasks. Prior works have proposed to learn fashion representation either by complex multi-task objectives or fine-tuning strong pretrained visual features such as CLIP. However, such methods fail to learn strong features that can distinguish between fashion-specific attributes/entities that are often available in image descriptions. Authors propose E^2 which add new layers over CLIP visual encoder along with selection tokens for different entities (such as brand, composition). They propose to attend these selection token with the image tokens such as each selection token can select a single image token. They use contrastive loss to supervise the network training and also use separate contrastive losses to supervise each of the selection tokens (supervision is provided from the categories). Authors have shown strong improvements over SOTA in image retrieval as well as zero-shot captioning task highlight the efficacy of the proposed representations. They also show some qualitative results to show the benefits of the proposed approach.

**Strengths:**

- The core idea of using selection tokens to attend to specific fashion attributes is interesting. Moreover, building it on top of CLIP can lead to better generation.
- The idea of gumbel-softmax with straight-through gradient trick for learning the hard attention is also interesting
- Authors have shown strong improvements over SOTA methods in both captioning and retrieval tasks. The ablation experiments also show improvement due to the proposed modules.
- Authors did provide additional details in the appendix such as detailed ablation studies, additional results on selection tokens

**Weaknesses:**

- Authors could have done a better job at presenting their approach and highlighting the core novelty (use of fusion blocks with selection tokens) in the introduction and abstract.
- Although the paper was easy to follow- the presentation could have been improved. For example the figures were often small to read.

**Questions:**

- Why did the authors use hard-attention for the selection tokens. Was there a difference in performance if doing soft-attention? The visualization of the selection tokens in figure14 is not showing much difference (the attention seems to be clustered around a few points). This leads to the question if the selection tokens are interpretable

- Why did the authors choose only one fashion dataset from prior works (other is proposed in this paper). Why weren't other datasets considered?

- What is the benefit of using >1 selection tokens for each sub-category?

- Would this work also generalize to other domains. If yes, what is needed to achieve this (e.g. detailed description + specific attributes)

**Limitations:**

Yes

---

> ### Author Rebuttal · Authors · 2024-08-03
>
> We sincerely thank you for your efforts in reviewing this paper and giving valuable suggestions!
>
> ## W1
>
> Thanks for the suggestion! In our current version, we have focused more on discussing the motivation and the benefits of our approach, including analytical discussions, in the Introduction and Abstract. Following your suggestion, we will introduce more technical novelty of our approach at the beginning in the final version.
>
> ## W2
>
> We appreciate your suggestion and apologize for the inconvenience caused by the small fonts due to limited space. In our final version, we will better arrange the format to allow for an improved layout, with better fonts in proper size.
>
> ## Q1
>
> We used the hard-attention because we wanted to explicitly select most relevant and informative image patch tokens while dropping less relevant ones, to enable the (merged) selection tokens containing **tag entity (or attributes)-specific knowledge**. In the Introduction, we show that such knowledge is critical to fashion tasks, and therefore we choose hard attention.
>
> Moreover, hard-attention could also lead to better performance (cross-modal retrieval on FashionGen with full-candidate evaluation):
>
>
> | Method          | I2T         |   I2T    |   I2T    | T2I    |  T2I   |  T2I   | SumR  | MeanR@1 |
> |-----------------|-----------------------|-------|-------|-----------------------|-------|-------|-------|---------|
> |                 | R@1                   | R@5   | R@10  | R@1                   | R@5   | R@10  |   -    |    -     |
> | Soft-attention  | 61.9                  | 88.5  | 94.8  | 63.4                  | 89.4  | 94.8  | 492.8 | 62.6    |
> | Hard-attention  | **62.8**              | **89.3** | **95.3** | **64.5**              | **90.1** | **95.5** | **497.9** | **63.7** |
>
> In Figure 14, we are not doing a comparison between hard-attention and soft-attention. Instead, we actually show: (1) the selected tokens in our model and (2) the tokens that have the maximal attention scores with the global token in the transformer in **vanilla CLIP-FT model**. For (2), the colored tokens (with high attention scores) are intuitively the ones that the CLIP-FT model gives more importance during inference.
>
> By the comparison, we want to show that our selected tokens can have a better coverage of important attributes in fashion domain.
>
> For example, we can find that:
>
> (1) Vanilla CLIP-FT tends to give more importance to image patch tokens of a solid color (or pure color) in fashion images. For instance, black areas in (g), (i), (j), and white areas in (k), (l).
>
> (2) By contrast, our selection tokens capture informative image patch tokens: E.g., 'Brand' tokens (blue) accurately capture the printed logo in (a), (b), (c), (d), (e) and (f).
>
> We have detailed explanation in Appendix B (Line 472).
>
> The quantitative and qualitative results demonstrate that our selection tokens are effective and interpretable.
>
> ## Q2
>
> In fact, in addition to FashionGen, we also included Fashion IQ [1] in Table 8, a fashion benchmark for the Text-guided Image Retrieval (TGIR) task. We evaluated the specific zero-shot TGIR task on this benchmark. We will add details of this dataset into the Section 4 (Datasets) for better clarity in our final paper.
>
> We evaluate the cross-modal retrieval and fashion captioning task with FashionGen because it is the most commonly used cross-modal retrieval benchmark in Fashion domain (e.g., it is used by ALBEF, SyncMask, FashionSAP, FashionViL, FashionBert, KaleidoBert, FaD-VLP, FAME-ViL, etc.). Both recent works (FashionSAP, SyncMask, FaD-VLP, FAME-ViL) and representative works (FashionBert, KaleidoBert) evaluate the cross-modal retrieval task with only FashionGen. We did not use other fashion datasets for cross-modal retrieval because there was not another widely-used and well-established benchmark for this task.
>
> ## Q3
>
> Intuitively, with more selection tokens for each sub-category, we could select more image patch tokens, and merge them with each selection token. In Appendix I, we analyzed how different numbers of selection tokens for each sub-category can affect model results in detail (quantitative results are available Table 10). Empirically, using 2 selection tokens for each sub-category achieves a good balance between accuracy and efficiency.
>
> ## Q4
>
> Conceptually "yes". As our model learns to attend to important attributes/regions with additional fusion blocks and selection tokens, our method could be applicable to other data/domains as long as specific attributes are provided.
>
> In fashion domain, the specific attributes are the existing tag entities, in other domains (such as medical images), we could directly use similar existing attributes (if any) and apply our model.
>
> If there are no such attributes in existing data immediately, one intuitive solution could be collecting useful attributes for existing data first (manually or using other models). For example, in contrastive learning for medical images [2,3,4,5], extracted entities [6] from image descriptions could be used as additional attributes.
>
> Finally, we would like to kindly remind that, fashion domain itself is an emerging topic [7-15], and our effective contrastive learning of fashion representations would benefit a series of downstream tasks such as cross-modal retrieval, fashion captioning, text-guided retrieval, recommendation, etc.
>
> Although our method also demonstrates insights in finetuning large-scale pre-trained vision-language models on any paired (image-text) data with additional attributes, and could be potentially applied to other domains (such as medical images), the primary focus of this work is still fashion representation learning [8,10,12,13,14,15].
>
> Thank you again for your thorough review of this paper and your valuable suggestions. We hope the responses could address your concerns. If you have any further questions, we look forward to having further discussions with you!
>
> # Reference
> (available in "Author Rebuttal by Authors")

---

> ### Author Response · Authors · 2024-08-14
>
> We sincerely thank you again for your efforts in reviewing this paper and giving valuable suggestions!
>
> Would you mind checking our responses to see if your previous concerns/questions have been addressed? We highly value your feedback, and if you have any more questions or suggestions, we look forward to having further discussions with you.
>
> Thanks!

---

### Author Rebuttal · Authors · 2024-08-03

## Reference

[1] Wu et al., Fashion IQ: A new dataset towards retrieving images by natural language feedback, CVPR 2021

[2] Irvin et al., Chexpert: A large chest radiograph dataset with uncertainty labels and expert comparison. AAAI 2019

[3] Johnson et al., Mimic-cxr, a de-identified publicly available database of chest radiographs with free-text reports. Scientific data

[4] Rahman et al., Exploring the effect of image enhancement techniques on covid-19 detection using chest x-ray images. Computers in biology and medicine, 2021

[5] Shih et al., Augmenting the national institutes of health chest radiograph dataset with expert annotations of possible pneumonia. Radiology: Artificial Intelligence, 2019

[6] Wang et al., MedCLIP: Contrastive Learning from Unpaired Medical Images and Text, EMNLP 2022

[7] Liu et al., Arbitrary Virtual Try-on Network: Characteristics Representation and Trade-off between Body and Clothing, ICLR 2023

[8] Bai et al., Cross-Domain Product Representation Learning for Rich-Content E-Commerce, ICCV 2023

[9] Baldrati et al., Multimodal Garment Designer: Human-Centric Latent Diffusion Models for Fashion Image Editing, ICCV 2023

[10] Pal et al., FashionNTM: Multi-turn Fashion Image Retrieval via Cascaded Memory, ICCV 2023

[11] Karras et al., DreamPose: Fashion Image-to-Video Synthesis via Stable Diffusion, ICCV 2023

[12] Han et al., FAME-ViL: Multi-Tasking Vision-Language Model for Heterogeneous Fashion Tasks, CVPR 2023

[13] Jiao et al., Learning Attribute and Class-Specific Representation Duet for Fine-grained Fashion Analysis, CVPR 2023

[14] Han et al., FashionSAP: Symbols and Attributes Prompt for Fine-grained Fashion Vision-Language Pre-training, CVPR 2023

[15] Song et al., SyncMask: Synchronized Attentional Masking for Fashion-centric Vision-Language Pretraining, CVPR 2024

[16] Wang et al., FashionVQA: A Domain-Specific Visual Question Answering System, CVPRW 2022.

[17] Cherti et al., Reproducible scaling laws for contrastive language-image learning, CVPR 2023

[18] Khosla et al., Supervised Contrastive Learning, NeurIPS 2020

---

### Decision · Program_Chairs · 2024-09-25

**Decision:**

Accept (poster)

**Comment:**

The reviewers generally appreciated the proposed approach, which is specifically designed for visual recognition in the fashion domain, where fine-grained visual representations are crucial for strong performance. They acknowledged that the core idea of using selection tokens to attend to specific fashion attributes is interesting and that the paper provides ample and convincing empirical evidence to support its effectiveness. While there was some concern about the limited scope of the paper's application to the fashion domain, the reviewers agreed that this alone should not be a reason for rejection, as the paper in its current state merits acceptance. After carefully reading the reviews and the rebuttal, this meta-reviewer concurs with the reviewers' unanimous recommendation for acceptance.